# Redox-Sensitive and Hyaluronic Acid-Functionalized Nanoparticles for Improving Breast Cancer Treatment by Cytoplasmic 17α-Methyltestosterone Delivery

**DOI:** 10.3390/molecules25051181

**Published:** 2020-03-05

**Authors:** Somayeh Rezaei, Soheila Kashanian, Yadollah Bahrami, Luis J. Cruz, Marjan Motiei

**Affiliations:** 1Department of Applied Chemistry, Faculty of Chemistry, Razi University, Kermanshah 6714414971, Iran; somayehrezaei128@gmail.com; 2Translational Nanobiomaterials and Imaging, department of Radiology, Leiden University Medical Centre (LUMC), 2333 ZA Leiden, The Netherlands; 3Nano Drug Delivery Research Center, Kermanshah University of Medical Sciences, Kermanshah 6734667149, Iran; 4Department of Medical Biotechnology, School of Medicine, College of Medicine and Public Health, Flinders University, Bedford Park, SA 5042, Australia; bahramiyadollah@yahoo.com; 5Department of Pharmacognosy & Pharmaceutical Biotechnology, Faculty of Pharmacy, Kermanshah University of Medical Sciences, Kermanshah 6714415153, Iran; 6Molecular Biology Research Centre, Kermanshah University of Medical Sciences, Kermanshah 6714415185, Iran; 7Centre of Polymer Systems, Tomas Bata University in Zlín, Třída Tomáše Bati 5678, 76001 Zlín, Czech Republic; marjanmotiei@yahoo.com

**Keywords:** reduction-responsive nanoparticles, hyaluronic acid, chitosan, lipoic acid, CD44, 17α-methyltestosterone

## Abstract

Novel reduction-responsive hyaluronic acid–chitosan–lipoic acid nanoparticles (HACSLA-NPs) were designed and synthesized for effective treatment of breast cancer by targeting Cluster of Differentiation 44 (CD44)-overexpressing cells and reduction-triggered 17α-Methyltestosterone (MT) release for systemic delivery. The effectiveness of these nanoparticles was investigated by different assays, including release rate, 3-(4,5-Dimethylthiazol-2-Yl)-2,5-Diphenyltetrazolium Bromide (MTT), lactate dehydrogenase (LDH), caspase-3 activity, Rhodamine 123 (RH-123), and Terminal deoxynucleotidyl transferase dUTP nick end labeling (TUNEL). In vitro experiments revealed that Methyltestosterone/Hyaluronic acid–chitosan–lipoic acid nanoparticles (MT/HACSLA-NPs) illustrated a sustained drug release in the absence of glutathione (GSH), while the presence of GSH led to fast MT release. HACSLA-NPs also showed high cellular internalization via CD44 receptors, quick drug release inside the cells, and amended cytotoxicity against positive CD44 BT-20 breast cancer cell line as opposed to negative CD44, Michigan Cancer Foundation-7 (MCF-7) cell line. These findings supported that these novel reduction-responsive NPs can be promising candidates for efficient targeted delivery of therapeutics in cancer therapy.

## 1. Introduction 

Chemotherapeutic agents inhibit tumor growth, angiogenesis and metastasis, and also increase the apoptosis of cancer cells through different mechanisms. Typically, these agents are used orally, and their administration is limited by poor aqueous solubility and bioavailability. Also, for patients who cannot take drugs via the oral route, poor aqueous solubility hinders their intravenous administration. The increase of the drug dose clinically to achieve the therapeutic concentrations at the target tissue can induce toxicity. Therefore, developing novel delivery approaches for these agents would be beneficial [1].

Todays, testosterone and its esters are considered important agents in the management of advanced mammary cancer. However, their application is restricted by the parental administration requirement. 17α-Methyltestosterone (MT) is a clinically effective androgen that can be administrated either via the oral or buccal cavity, with a different effect from parenterally administered testosterone or its esters on excretion of hormones, carbohydrate metabolism, production of gynecomastia and the function of the liver [2]. MT is utilized during menopause as a supplement to typical hormone substitution therapy. This substance with a weak dependence on the androgen receptor is an orally active synthetic androgen, which is currently the only androgen in the management of estrogen–androgen Hormone Replacement Therapy (HRT) in the United States with Food and Drug Administration (FDA)-approval [3,4,5]. Studies suggest that dwarfed children and adults with metastatic breast carcinoma subjected to MT showed a reduction in the protein-bound iodine of serum and also in the uptake of radioiodine [6].

Nanoparticles (NPs) are vehicles that can prolong drug circulation time, transfer hydrophobic drugs, enhance the accumulation of drugs in tumor tissues through enhanced permeability and retention effect, and reduce normal tissue toxicity. Therefore, various kinds of NPs, including liposomes, micelles, and magnetic NPs, were fabricated to enhance the therapeutic effects of anticancer drugs [7,8,9]. However, inefficient drug release of the traditional micelles at the tumor site restricts the therapeutic outcomes. Besides, weak stability of the existing micellar drug formulations in the circulation leads to the leakage of drugs. Therefore, numerous studies have been accomplished to develop more stable polymeric micelles and to have more efficient drug release in tumor tissues [10].

Now, nanomaterials, which respond to glutathione (GSH), are of great interest due to their enhanced drug release in the cytoplasm [11]. Redox stimulus-responsive drug delivery systems have been recently developed to resolve the above-mentioned challenges. In this method, intracellular redox potential results in a burst release of encapsulated drugs because of the structure of the NPs containing disulfide bonds [12]. GSH tripeptide is a biological reducing agent with a low molecular weight and high redundancy diminished by nicotinamide adenine dinucleotide phosphate (NADPH) and GSH reductase. GSH is more concentrated in the cytosol and cell nucleus compared to the body fluids and extracellular matrices (100–1000 times higher); the former has values between 2 to 10 mM, while the latter has 2–10 M [12,13]. Redox-sensitive hyaluronic acid–chitosan–lipoic acid (HACSLA) conjugates may enhance the therapeutic efficacy by triggering tumor-specific intracellular drug release since there is a considerable difference between intra and extracellular environments in the redox potential.

Hyaluronic acid (HA), a biocompatible, biodegradable, and non-antigenic polysaccharide [9], has various applications as a conjugate, nanogel, and hydrogel in biomedicine [14]. Having potential for active targeting is the most attractive property of HA in nanomedicine. CD44 receptor and Receptor for Hyaluronan Mediated Motility (RHAMM) on the surface of cancer cells with a high affinity to HA has been demonstrated [15,16]. CD44 is a transmembrane glycoprotein overexpressed in various types of tumor cells, including breast, colorectal, lung, and melanoma cancer cells. HA can specifically attach to CD44. The unique advantages of HA, including modification flexibility, non-toxicity, and non-immunotoxicity, make it different from other kinds of targeting moieties [17]. A great number of methods, including synthesis of the HA-drug conjugate, HA grafted copolymer, HA-coated the polymeric NPs, hydrophobic associated HA-based soft nanogel carriers, and ionotropic gelation (IG), have been expanded to fabricate the HA-based drug delivery systems [18]. Chitosan (CS) as a biocompatible, biodegradable, and nontoxic polymer has been under various investigations for the fabrication of the HA-coated NPs using the IG method. [18,19]. Thereafter, MT was encapsulated into the hydrophobic core of the nanoparticle containing a disulfide bond of Lipoic acid (LA) utilized as a reduction part [20].

In the present work, we investigated the physicochemical and then biological properties of NPs to exhibit high drug-loading capacities for MT through redox-sensitive Chitosan-Lipoic acid (CSLA) conjugates or insensitive HACSLA conjugates. First, drug-loaded NPs were evaluated in terms of any alteration in morphology and MT state in different reducing environments. In addition, BT-20 cancer cell line with overexpression of CD44 and MCF-7 cancer cell line with low CD44 expression were used to confirm the selective binding affinity of HACSLA NPs to the HA receptors [21,22]. In vitro MTT, lactate dehydrogenase (LDH), Caspase-3 activity, Terminal deoxynucleotidyl transferase dUTP nick end labeling (TUNEL), and Rhodamine 123 (RH-123) assays were also accomplished to evaluate tumor-specific cytotoxicity and apoptosis induction due to MT-loaded HACSLA-NPs.

## 2. Results 

### 2.1. FT-IR and ^1^H NMR Techniques for Confirmation of HACSLA-NPs Synthesis

Figure 1 shows the Fourier transform infrared spectroscopy (FT-IR) spectra of CS, LA, CSLA-NPs, HA, HACSLA-NPs, MT, MT/CSLA-NPs, and MT/HACSLA-NPs. When CS and LA peaks are compared with CSLA-NPs, the changes can be ascribed to the formation of the amidic bond between CS and LA. The peak at 1670 cm^−1^ (C=O stretching) is characteristic of the amide group. The spectra of MT loading CSLA-NPs were compared to MT (Figure 1d) and CSLA-NPs (Figure 1c), respectively. Thus, the FT-IR spectra of the reaction product are similar to the suggested structural formulae presented in (Figure 1e). Therefore, the bands correspond to the stretching vibration at 3450 cm^−1^ of O–H from MT, and the disappearance of the stretching vibration at 1664 cm^−1^ corresponds to the C=O bond from MT [23]. The construction of HA-coated CSLA-NPs was investigated using IR spectroscopy, as shown in (Figure 1g). It was observed that the characteristic absorption peaks of CSLA-NPs and HA were all visible along the spectrum of the HA-coated CSLA-NPs. This confirmed that CSLA-NPs and HA were involved in HACSLA-NPs formation. Compared with the spectra of CSLA-NPs and HA (Figure 1f), there were no obvious new absorption peaks in the spectrum of HA-coated CSLA-NPs, thus indicating that no new bonds were formed during the preparation of HA-coated CSLA-NPs. HA was absorbed to the surface of the CSLA-NPs via electrostatic interactions between CS and HA, rather than by chemical conjugation [24]. CSLA-NPs were prepared via the acylation reaction between the CS and LA [20]. HA and CSLA-NPs were formed via electrostatic interactions and purified by dialysis method and filtration.

CSLA-NPs and HACSLA-NPs were also characterized by proton nuclear magnetic resonance (^1^HNMR), as shown in Figure 2. The ^1^H NMR spectrum of CS showed virtually three sets of signals at 4.67 ppm, 3.20–4.00 ppm, and 2.01 ppm attributed to H1, sugar ring, and –CH2, respectively. The successful LA conjugated CS was proved by specific proton peaks of penta-heterocyclic structure in LA at 2.31 ppm, 2.67 ppm, and 3.18 ppm [25]. The introduction of HA to HACSLA-NPs was also confirmed by the presence of two peaks at 7.70 ppm and 6.84 ppm. All the above-mentioned results recommended the successful synthesis of CSLA-NPs and HACSLA-NPs.

### 2.2. Size and Surface Charge

Dynamic light scattering (DLS) analysis showed a relatively narrow hydrodynamic diameter of 240 ± 0.056 nm in CSLA-NPs, as confirmed by low polydispersity index (PDI) values of 0.369 ± 0.056. Laser Doppler velocimetry (LDV) analysis also demonstrated the zeta potential of CSLA-NPs up to +26 mV. After the addition of HA, the size of HACSLA-NPs was increased to 280 ± 0.045 nm with PDI values of 0.327 ± 0.002, and the zeta potential decreased to +19 mV. The positive charge of NPs was related to unbounded H of CS, but less zeta potential of HACSLA-NPs was related to the negative charge of HA, so the size and surface charge of the NPs were changed significantly, from 240 to 280 nm and 26.0 to 19 mV, respectively.

### 2.3. Morphology

Scanning electron microscope (SEM) images of HACSLA-NPs (Figure 3a) indicate orderly spherical shape with no accumulation and distribution of the prepared NPs. HACSLA-NPs have the size around 50 and 130 nm in diameter, respectively. Energy dispersive X-ray microanalysis (EDX) was performed for the qualitative analysis of CSLA-NPs. Elemental analysis of the NPs was carried out using EDX in SEM. The EDX spectrum of the NPs is shown in Figure 3b. The vertical and horizontal axis shows the number of x-ray counts and energy (Kev), respectively. The EDX analysis was performed to investigate the elemental composition of the NPs, and to ensure the presence of S as an indication for the presence of LA embedded in the NPs. The EDX spectrum indicated various intensive peaks that are associated with H, N, O, and S atoms for the samples, in which the presence of S peak is completely obvious.

### 2.4. X-ray Powder Diffraction (XRD)

To realize the nature of MT after being encapsulated into CSLA-NPs, the X-ray powder diffraction (XRD) method was used. The diffraction patterns of the MT, HA-CSLA NPs, and MT/HACSLA-NPs are shown in Figure 4. The pure MT exhibited sharp peaks in the range of 14.48, 15.72, 16.78 and 19.82, which implied a high crystalline structure (Figure 4a) [26]. Nevertheless, this characteristic was not observed in the MT/HACSLA-NPs (Figure 4c), which indicates that MT entrapped in the core of HACSLA-NPs in the amorphous or disordered-crystalline phase. In addition, there was not much difference in the diffraction pattern between HACSLA-NPs (Figure 4b) and MT/HACSLA-NPs, which suggests that the addition of MT did not change the nature of the HACSLA-NPs [26,27].

### 2.5. MT Release from NPs at Different pH Values and Conditions

The tumor microenvironment could always play the main role in controlling the release of anticancer drugs because GSH content or pH value of cancer cells differ from healthy cells. Therefore, the above-mentioned differences could avoid drug leakage and its burst release to enhance drug delivery and therapeutic efficacy [28]. Loading of MT into CSLA-NPs and HACSLA-NPs was performed at a theoretical drug loading (DL) of 10.30 *wt*% and a polymer concentration of 2 mg/mL. MT-loaded NPs showed the highest MT loading efficiency up to 96.59%.

The in vitro drug release studies were also performed on CSLA-NPs and HACSLA-NPs at 37 °C, pH of 7.4 (Figure 5a), and 6.5 (Figure 5b) and also at the concentration of 2 mg/mL. The results showed that the release of MT from CSLA-NPs and HACSLA-NPs at a pH of 7.4 and 6.5 without GSH was largely inhibited, in which only about 31.33, 31.73% and 34.75, 20.56% of the drug were released in 24–120 h from CSLA-NPs and HACSLA-NPs (Figure 5a,b). Nonetheless, 10 mM of GSH induced significantly the drug release up to 100, 73.30% at pH 7.4 and 55.15, 57.32% at pH 6.5 from CSLA-NPs and HACSLA-NPs, respectively. That was due to the opened lipoyl rings of the NPs core under the GSH reduction, which led to the swelling of the MT-loaded NPs and triggered drug release. Under 10 mM concentration of GSH and at pH values of 7.4 and 6.5, the release of MT from non-pH- sensitive CSLA-NPs and HACSLA-NPs was also examined. According to the results, CSLA-NPs and HACSLA-NPs did not show any pH-sensitive manner. CSLA-NPs and HACSLA-NPs released an insignificant amount of MT in the circulatory system as opposed to cancer cells. In addition to lower systemic toxicity, this manner might facilitate an enhanced anti-tumor effect.

### 2.6. Biological Assays

#### 2.6.1. MTT Assay

The cell viability data by MTT assay are shown in (Figure 6a,b). Different samples inhibited cell growth in a dose-dependent manner. For MCF-7 cells MT/CSLA-NPs showed strongest inhibition compared to MT/HACSLA-NPs and for BT-20 cells MT/HACSLA-NPs > MT/CSLA-NPs. These results support that both blank CSLA-NPs an HACSLA-NPs were nontoxic toward MCF-7 and BT-20 cells up to the tested NPs concentration of 2 mg/mL [29,30]. The MT had IC_50_ of 40.85 ± 0.904 µM and 47.58 ± 0.792 µM in MCF-7 and BT-20 cells, respectively. MT/CSLA-NPs enhanced the sensitivity of MCF-7 cell lines 1.3 fold lower than free MT (31.27 ± 0.547 µM comparing to 40.85 ± 0.904 µM). However, in BT-20 cells, the IC50 of MT/HACSLA-NPs (32.59 ± 0.538 µM) was lower than MT/CSLA-NPs (79.47 ± 1.02 µM). These results showed that HACSLA-NPs have the ability to target CD44-positive cells and then deliver and release MT into them.

#### 2.6.2. LDH Assay

The cytotoxicity data of samples proved our assumption about cell viability data. In MCF-7 cells (Figure 7a), after 48 h, the cell cytotoxicity increased in a dose-dependent manner by MT/CSLA-NPs compared to MT/HACSLA-NPs. By contrast, in BT-20 cells (Figure 7b), the cytotoxicity of MT/HACSLA-NPs was significantly high. LDH and MTT assay confirmed that CSLA-NPs and HACSLA-NPs showed no toxicity toward MCF-7 and BT-20 cells, and the integrity of the cell membrane is mostly preserved. As mentioned before in BT-20 cells, MT/HACSLA-NPs exhibited increase internalization via CD44 receptor compare to MT/CSLA-NPs, but in CD44 negative MCF-7 cell lines, both NPs can only be internalized via endocytosis [29] by means of other factors including size, and morphology. It should be mentioned that smaller NPs show greater cytotoxicity via induction of oxidative stress [1]. Therefore, MT/CSLA indicates more toxicity comparing to MT/HACSLA.

#### 2.6.3. TUNEL Assay

TUNEL can detect the early stages of DNA fragmentation in apoptotic cells [31]. The tunnel staining assay by evaluation the mean cell death index showed a significant difference between the various concentrations of MT and the control groups (*p* < 0.05). TUNEL data exhibited the detectable apoptosis by MT /CSLA-NPs in MCF-7 and MT/HACSLA-NPs in BT-20 as opposed to the unloaded NPs in both cell lines. The data were in agreement with MTT and LDH data (Figure 8).

#### 2.6.4. Caspase-3 Assay

Due to the central role of caspase-3 in cell death, activation of caspase-3 will increase the cytotoxicity effects and then apoptosis [32]. In most groups, apoptosis improved by increasing the concentration of MT (Figure 9). According to the data, the unloaded NPs did not show any caspase-3 activity. There was increased caspase-3 activity in MT- loaded NPs as compared to free MT in both MCF-7 and BT-20 cell lines. Caspase-3 activation of MT-loaded-NPs in MCF-7 was in the order of MT/CSLA-NPs > MT/HACSLA-NPs and for BT-20 MT/HACSLA-NPs > MT/CSLA-NPs.

#### 2.6.5. Rhodamine 123 (RH-123) Assay

We used RH-123 assay to specifically localize mitochondria in living cells and dead cells [33]. According to caspase-3 activity results, it was expected that MT, MT/CSLA-NPs, and MT/HACSLA-NPs ultimately trigger an apoptotic pathway after changing the mitochondrial membrane potential (Δφm). To check the changing of Δφm in the treated cells, cells were exposed to various concentrations of MT, MT/CSLA-NPs, and MT/HACSLA-NPs. Then, Δφm was measured by RH-123 staining and colorimetric assay after 24h (Figure 10). These data demonstrate that MCF-7 and BT-20 cells poorly accumulate RH-123 after treatment with MT- loaded NPs; by contrast, both cells showed a high RH-123 level in blank NPs.

#### 2.6.6. Immunocytochemical Staining of CD44

Immunohistochemistry can be used to evaluate the expression and subcellular localization of proteins and other molecules in tissues. It can be a perfect tool in cancer therapy, where it is used as a method to verify the identity of tissue types, to categorize tumors, and to estimate the presence of specific molecules for different applications such as therapeutic or prognostic goals [21]. Herein, We used Immunohistochemistry to characterize CD44 expression levels on the surface of MCF-7 and BT-20 breast cancer cells, as shown in (Figure 11b) proved that MCF-7 was negative CD44, while BT-20 showed positive CD44 [22]. Our results (Figure 11b) also confirmed a high expression of CD44 in BT-20 breast cancer cell line, while the CD44 expression was not detectable in the MCF-7 cell line.

## 3. Materials and Methods

### 3.1. Materials

CS (degree of deacetylation (DDA) 90.28% and molecular weight (MW) 50–200 kDa) was purchased from MP Biomedicals (Eschwege, Germany), HA, R-Alpha-LA, 1-Ethyl-3-(3-dimethylaminopropyl) carbodiimide (EDC)/N-hydroxysuccinimide (NHS), GSH, Tween 20, Trypsin/EDTA solution 0.25%, rabbit polyclonal antibodies (Abs): CD44, and MTT were provided from Sigma–Aldrich (St. Louis, MO, USA). Fetal Bovine Serum (FBS), L-glutamine (200 mM), and penicillin-streptomycin (10000 U/mL) were obtained from Biochrom (Berlin, Germany). Dead-end Fluorometric TUNEL System and Cell Titer 96^®^ Aqueous One Solution Cell Proliferation Assay were prepared from Promega (Madison, USA). MCF-7 and BT-20 cell lines were supplied from Pasteur Institute (Tehran, Iran) and MT from Aburaihan Pharmaceutical (Tehran, Iran). Dulbecco’s Modified Eagle Medium (DMEM) and MEM media, Fluorescein isothiocyanate (FITC)-conjugated (green) secondary Ab, and anti-fade reagent were obtained from GIBCO Invitrogen (Grand Island, NY). Kits of LDH cytotoxicity, Caspase-3 Fluorescence assay, and JC-1 Mitochondrial Membrane Potential assay were supplied from (Cayman Chemical, Michigan, MI, USA). All of the other chemicals and solvents were purchased from Merck (Darmstadt, Germany).

### 3.2. Synthesis of the Chitosan-Lipoic Acid Nanoparticles (CSLA-NPs)

The CSLA-NPs were produced by an amidation reaction. LA was conjugated onto CS using EDC/NHS. First, CS (0.01 g) was dissolved in 5 mL of acetic acid (2% *v/v*) under bath sonication for 90 min. Thereafter, an equal amount of EDC and NHS (0.05 g) were added to the solution of LA (0.05 g) in ethanol under stirring at 45 °C for 15 min. The amide bonds were formed after addition of CS solution into the LA solution under stirring at 45 °C for 12 h. Afterwards, dialysis tube (12 kDa MWCO) was utilized for 24 h against dH_2_O and finally the freeze-dried NPs were obtained [20].

### 3.3. Preparation of MT-Loaded CSLA-NPs

The dialysis method was used for the production of MT-loaded NPs. For this purpose, 1 mL MT solution (3.05 mg/mL) was added to 7 mL of blank CSLA-NPs (2 mg/mL) described in Section 3.2. MT-loaded NPs were dialyzed against dH_2_O for adequate time (8 h) to remove the free MT. Then, the entrapment efficiency (EE) and DL were calculated by the following formulas [34].

(1)$$\mathrm{EE}\;(\%)\;=\;\frac{Wa}{Wt}\times 100\%$$

(2)$$\mathrm{DL}\;(\%)\;=\;\frac{Wa}{Wt}\times 100\%$$

Wa, Wt, and W are the weight of MT loaded in the nanoparticle, the total weight of MT added into the system, and nanoparticle weight, respectively.

### 3.4. Synthesis of the MT/HACSLA-NPs 

MT/HACSLA-NPs blend was prepared by previously reported methods [35,36]. NPs were created by ionotropic gelation interaction. CS with opposite charge were cross-linked with HA [37]. For this purpose, MT/CSLA-NPs and HA were mixed in the ratio 2:1 without the use of a cross-linking agent. 7 mL of MT/CSLA solution was mixed into HA (13.3 mg/mL) solution and stirred for 10 min. Thereafter, MT/HACSLA-NPs were characterized by Fourier transform infrared spectroscopy (FT-IR) (Bruker, Bremen, Germany) and ^1^H NMR (Bruker, Bremen, Germany).

### 3.5. Particle Size and Zeta Potential

Hydrodynamic diameter and zeta potential of the NPs were analyzed by dynamic light scattering (DLS) and laser Doppler velocimetry (LDV) (Zetasizer, Malvern Nano ZS, Malvern Instruments, (Worcestershire, UK), respectively. 

### 3.6. Morphology

SEM ((FE-SEM) Field Emission Scanning Electron Microscope: TESCAN BRNO-Mira3 LMU, Brno, Czech Republic) was used for the determination of the morphology characteristics of HACSLA-NPs at an operating voltage of 25 kV. An Energy Dispersive X-Ray Analyzer (EDX) (Brno, Czech Republic) was also used to provide elemental (H, O, C, S) identification and quantitative NPs information.

### 3.7. XRD

The crystalline/amorphous structure of NPs was performed using a powder X-ray diffractometer (PANalytical, X’PertPro, and the Netherlands) with a copper anode (CuKa radiation). The instrument was set at 40 kV and 30 mA, and the diffraction angle was set from 0 < 2 > 80 at a step size of 0.05 and step time of 1 s. XRD patterns were obtained with DFFRAC plus X-Ray Diffraction Software (XRD Laboratory — University of Kashan) having an XRD commander program.

### 3.8. In Vitro Release of MT-Loaded CSLA-NPs and MT-Loaded HACSLANPs

The release profiles of MT loaded CSLA-NPs and HACSLA-NPs were studied by the dialysis bag diffusion technique at 37 °C in different release media, including Phosphate Buffered Saline (PBS) (pH 7.4), GSH (10 mM), PBS (pH 6.5), GSH (10 mM). The MT-loaded CSLA and HACSLA NPs were poured into the dialysis tube (3.5 kDa MWCO), then they were soaked in 40 mL of the release media under shaking conditions at a speed of 100 rpm at 37 °C. Thereafter, the released MT was evaluated by UV–vis spectrophotometry (Cecil Instruments Limited, Model: CE 1021, Cambridge, UK) technique at 241 nm versus a calibration curve [20].

### 3.9. In Vitro Cellular Studies

#### 3.9.1. Cell Culture

DMEM and MEM media were used for culturing MCF-7 and BT-20 cell lines, respectively [38,39]. These media were supplemented with FBS (10%), penicillin (100 U/mL), and streptomycin (100 mg/mL) and incubated at 37 °C in a humidified conditions in an atmosphere of 5% CO2.

#### 3.9.2. MTT Assay

MTT assay was utilized for cell viability evaluation. 15 × 10^3^ cells were seeded in 96-well plate for 24 h in the presence of 200 μL of media containing 10% FBS. Thereafter, MCF-7 and BT-20 cells were subjected to the various samples ((a) MT, (b) CSLA-NPs, (c) MT-loaded CSLA-NPs, (d) HACSLA-NPs and (e) MT-loaded HACSLA-NPs) containing different concentration of MT (0.0–60 μM) for 24 h. Thereafter, each well was exposed to 50 μL of MTT solution (5 mg/mL) for 3 h and then 100 μL of dimethyl sulfoxide (DMSO). Finally, the optical density of each well was determined by an ELISA reader at 570 and 630 nm [40].

#### 3.9.3. LDH Assay

LDH assay was also used for the determination of cell toxicity because the damaged cells would release LDH into the media. Therefore, the increased amount of LDH in each media indicates that the solutions have shown cytotoxicity effects on cells [41]. For LDH assay, 2 × 10^5^ cells/well were seeded in 24-well plate for 12 h. After exposure to the samples for 24 h, conditioned media of cells were collected and added to a fresh 96-well plate containing LDH assay reagent. The colorimetric assay of LDH activity was performed using an Enzyme Linked Immunosorbent Assay (ELISA) reader at 490 nm after incubation for 30 min.

#### 3.9.4. Caspase-3 Activity Assay

MCF-7 and BT-20 cells were seeded in a 96-well plate at density15 × 10^3^ for 24, and 48 h, treated with different concentrations of MT (i.e., 5, 10, 20, 30 and 40 µM) as five groups including (a), (b), (c), (d) and (d) described at section (2.9.1.) and then incubated for 24 and 48 h. Finally, cleavage of the caspase-3 substrate (N-acetyl-DEVD-p-nitroaniline) [42] was monitored by measuring the fluorescent density at emission and excitation wavelengths of 535 nm and 485 nm, respectively. Data were obtained from two independent experiments.

#### 3.9.5. Rhodamine 123 (RH-123) Assay

For quantitative analysis, mitochondria membrane potential (MMP) was measured using the cell-permeable cationic fluorescent probe RH-123. Briefly, MCF-7 and BT-20 cells (3 × 10^4^ cells/well) were cultured and treated in different MT solutions. After washing with PBS, they were incubated by 1 μM RH-123 in a dark room at 37 °C for 30 min. Finally, the absorbance of the cells was measured at specified excitation (488 nm) and emission (525 nm) wavelengths using an ELISA Reader [43]. The reference wavelengths should be more than 630 nm. Experiments were repeated independently at least 3 times.

#### 3.9.6. TUNEL Assay

A dead-end fluorometric TUNEL System was used according to the manufacturer’s protocol to identify the apoptotic cells. Briefly, after seeding the cells in 96-well plates and treatment with the five samples, 4% *w/v* paraformaldehyde in PBS was utilized for 10 min at 4 °C to fix the cells. Thereafter, they were permeabilized by ice-cold Triton-x100 (2%) for 2 min, and incubated with the reagent to catalyze the polymerization of labeled nucleotides to 3′OH terminals of DNA fragments. After addition of propidium iodide (PI), the treated cells were observed by a fluorescent microscope (Olympus AX-70, Missouri City, TX, USA) [44].

#### 3.9.7. Immunocytochemical Staining of CD44

MCF-7 and BT-20 cell lines were seeded in 96-well culture plates with the density of 10^4^ cells/well for 24 h, fixed in 4% *w/v* paraformaldehyde for 20 min at 25 °C, and then permeabilized in Triton/PBS (0.1%) for 5 min. To decrease nonspecific binding, the cells were incubated with PBS containing FBS (0.5%) and Tween 20 (0.1%) for 30 min. After that, the cells were incubated overnight with the rabbit polyclonal Abs: CD44 (1:60) at 4 °C. After washing, FITC-conjugated (green) secondary Ab (1:100) was used for 1 h at 25 °C. Finally, the cells were treated with an anti-fade reagent and analyzed by a fluorescent microscope (Olympus AX-70) [45].

### 3.10. Statistical Analysis

Data were analyzed using Excel program version 2013 were used to analyzing data. All results were expressed as mean ± standard deviation (SD). One-way analysis of variance (ANOVA) followed by *t*-test was used to parallel the differences among the means and comparison of data using SPSS statistical software, version 21 (SPSS Inc., Chicago, IL, USA). A value of *p* < 0.05 was considered statistically significant.

## 4. Conclusions

In this work, we successfully synthesized CSLA-NPs and HACSLA-NPs as drug delivery systems with effective intracellular MT release and improved cellular internalization. The decoration of the cationic CSLA-NPs with HA changed their size from 240 ± 0.056 to 280 ± 0.045 nm and Zeta potential from 24.0 to 19.0 mV. The larger size of the HA-coated NPs is indicative of the HA layer on the surface [46], and the decrease in Zeta potential of HACSLA-NPs is due to the reduced protonation of CS. It can be a cause of amending HACSLA-NPs’ stability in blood circulation and reducing the non-specific interaction with serum proteins in comparison with CSLA-NPs [20]. Then, we have compared the in vitro performance of the HA-coated and uncoated NPs. HA-coated NPs reduced MT release at pH 7.4 and increased MT release at pH 6.5. Therefore, HACSLA-NPs improved selective targeting against cancer cells by a faster drug release profile at pH 6.5, which simulates the cytoplasm environment of the targeted tissue with an improved therapeutic efficacy. We have also confirmed highly efficient and targeted therapy of breast cancer cell lines (i.e., BT-20 and MCF-7) by MT/CSLA-NPs and MT/HACSLA-NPs using cytotoxicity and apoptosis assays, which confirmed that MT/HACSLA-NPs exhibited more detectable cytotoxicity and apoptosis effect on positive CD44 BT-20 cells than MT/CSLA-NPs. MT/CSLA-NPs and MT/HACSLA-NPs have several features such as small size, spherical morphology and fast GSH-responsive drug release, in addition to higher selectivity toward CD44 receptors by MT/HACSLA-NPs. It can be postulated that the presence of HA on the NPs’ surface will affect not only on MT release rate at different pH values but also on their preferential internalization in cancer cells overexpressing HA receptors. Thereafter, we have investigated the cellular responses of MCF-7 and BT-20 cell lines to unloaded and MT-loaded NPs at varying doses. The results confirmed that MT-loaded NPs would damage the plasma and mitochondrial membranes, which can be attributed to LDH release into the extracellular medium. In summary, our results proved that HA surface modification played a major role in the biological responses, and these reduction-sensitive NPs can be utilized effectively in biomedical applications.

## Figures and Tables

**Figure 1 molecules-25-01181-f001:**
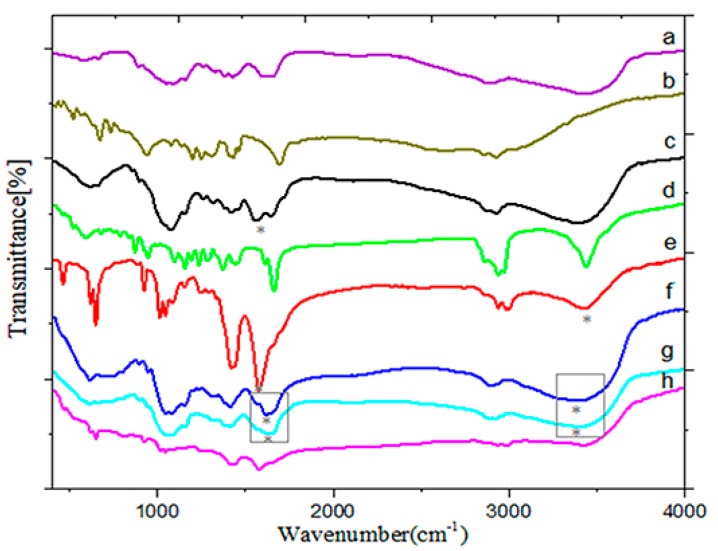
Fourier transform infrared spectra (FT-IR) of (**a**) Chitosan (CS), (**b**) Lipoic acid (LA), (**c**) Chitosan–lipoic acid nanoparticles (CSLA-NPs), (**d**) 17α-Methyltestosterone (MT), (**e**) 17α-Methyltestosterone/Chitosan–lipoic acid nanoparticles (MT/CSLA-NPs), (**f**) Hyaluronic acid (HA), (**g**) Hyaluronic acid–chitosan–lipoic acid nanoparticles (HACSLA-NPs), (**h**) 17α-Methyltestosterone/Hyaluronic acid–chitosan–lipoic acid nanoparticles (MT/HACSLA-NPs) (The peaks which are discussed in the text were marked by stars).

**Figure 2 molecules-25-01181-f002:**
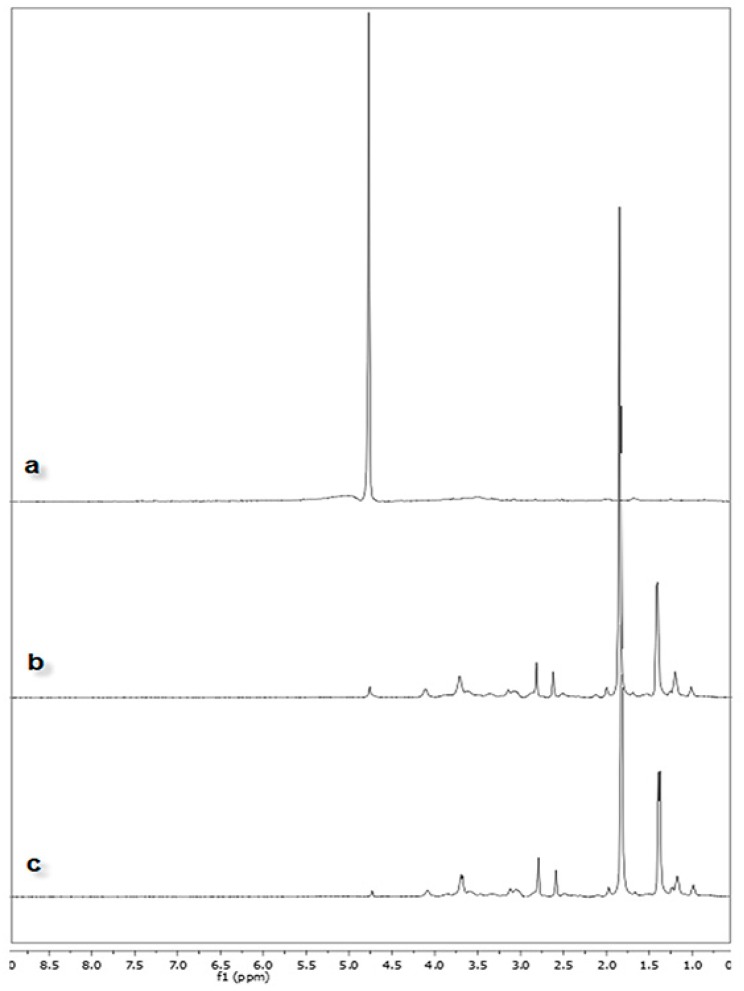
Proton nuclear magnetic resonance (^1^HNMR) spectrum of in Deuterium Oxide (D_2_O) of (**a**) CS, (**b**) CSLA-NPs and (**c**) HACSLA-NPs.

**Figure 3 molecules-25-01181-f003:**
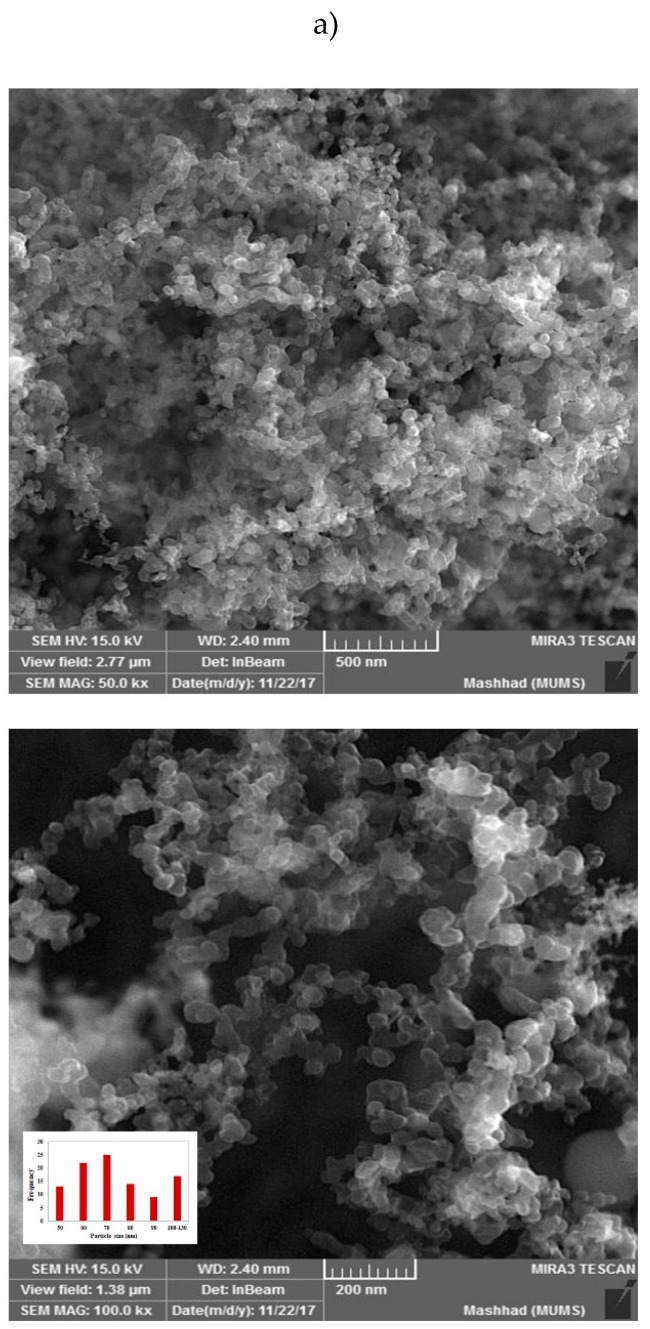
The size distribution and scanning electron microscope (SEM) images (**a**) with energy dispersive X-ray microanalysis (EDX) (**b**) analysis of HACSLA-NPs.

**Figure 4 molecules-25-01181-f004:**
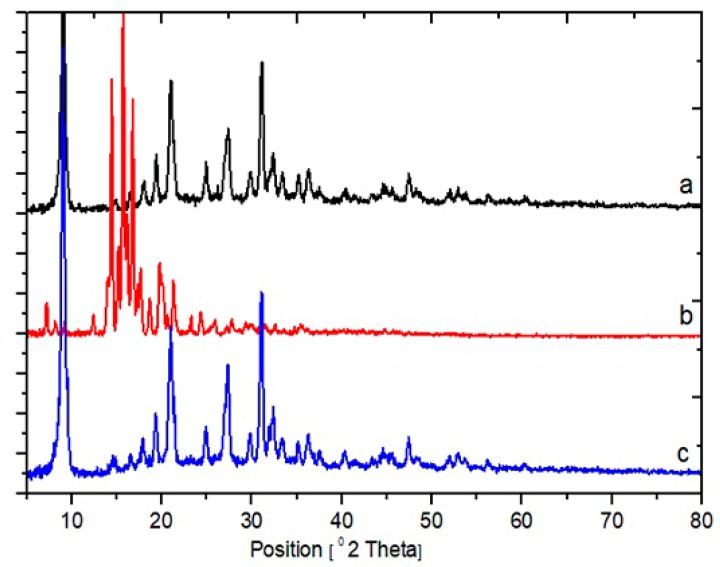
X-ray diffractograms of (**a**) MT, (**b**) HACSLA-NPs and (**c**) MT/HACSLA-NPs.

**Figure 5 molecules-25-01181-f005:**
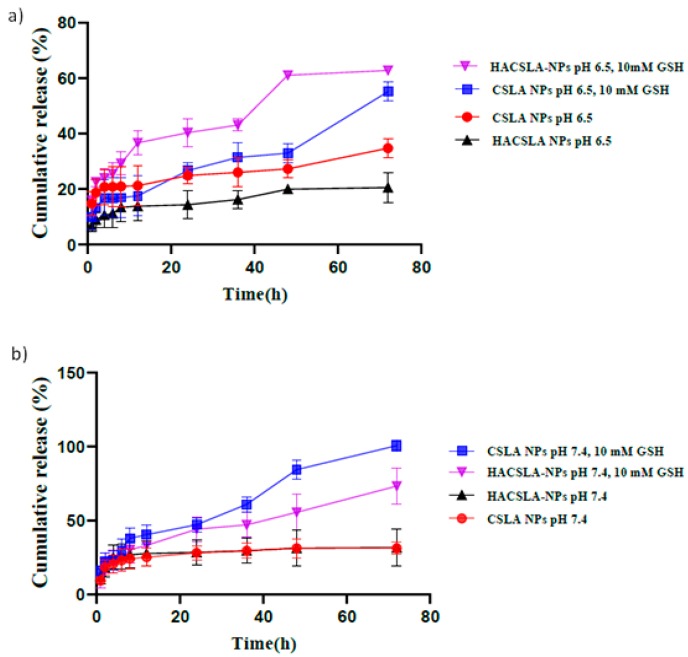
In vitro MT release profiles from (**a**) MT/CSLA-NPs, MT/HACSLA-NPs at pH 7.4 and (**b**) MT/CSLA-NPs, MT/HACSLA-NPs at pH 6.5 values and reduction conditions.

**Figure 6 molecules-25-01181-f006:**
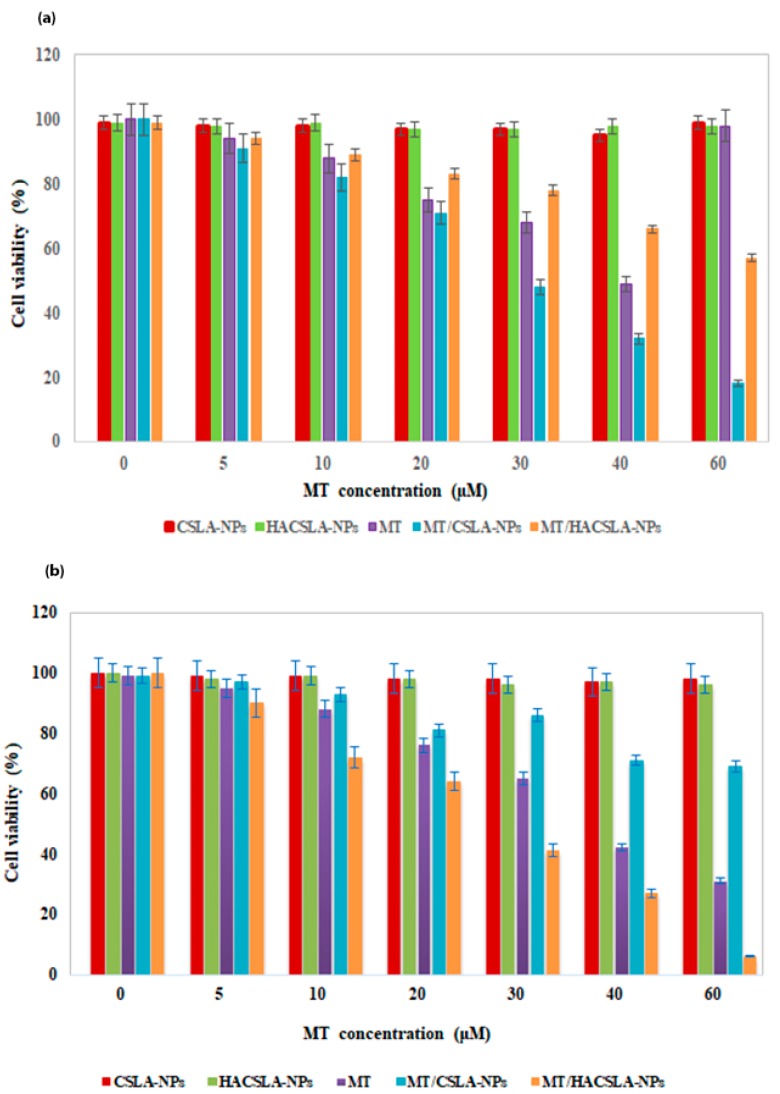
The effects of different MT concentrations (0.0–60 μM) on the cell viability on (**a**) Michigan Cancer Foundation-7 (MCF-7) and (**b**) BT-20 cells. All data represented by mean ± S.E.M (*p* < 0.05).

**Figure 7 molecules-25-01181-f007:**
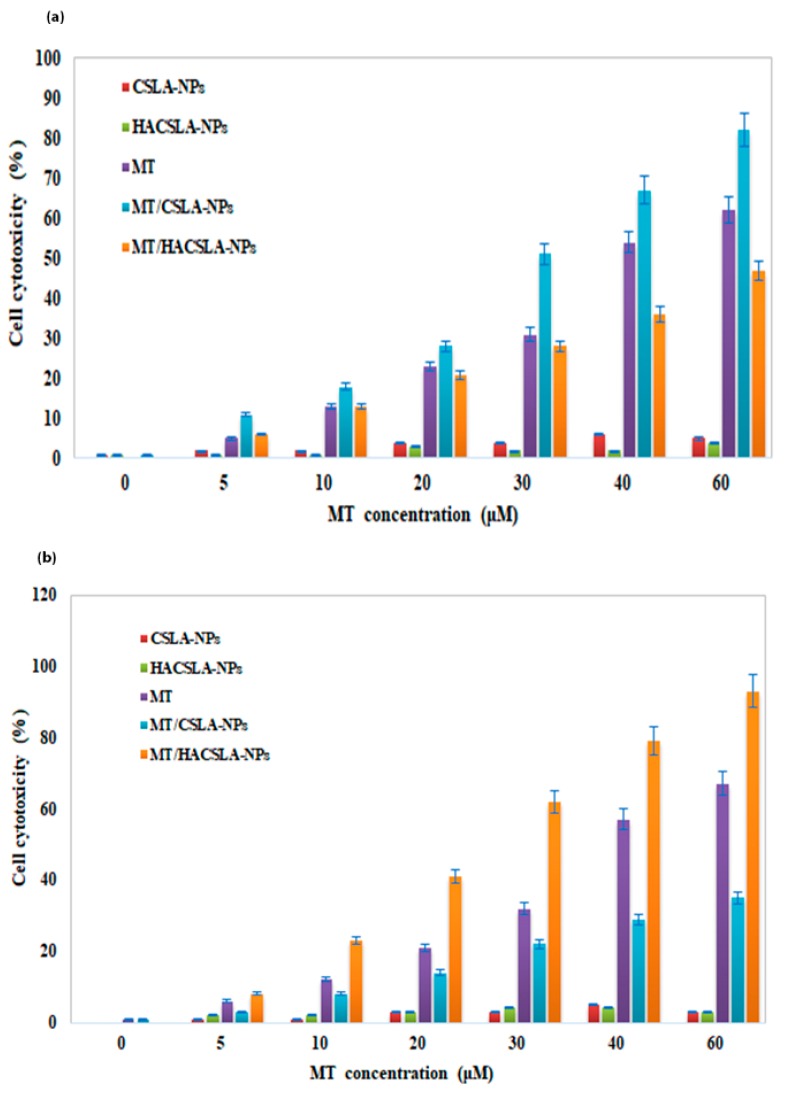
MT cytotoxicity measured with the LDH assay in (**a**) MCF-7 and (**b**) BT-20 cells. All data represented by mean ± S.E.M (*p* < 0.05).

**Figure 8 molecules-25-01181-f008:**
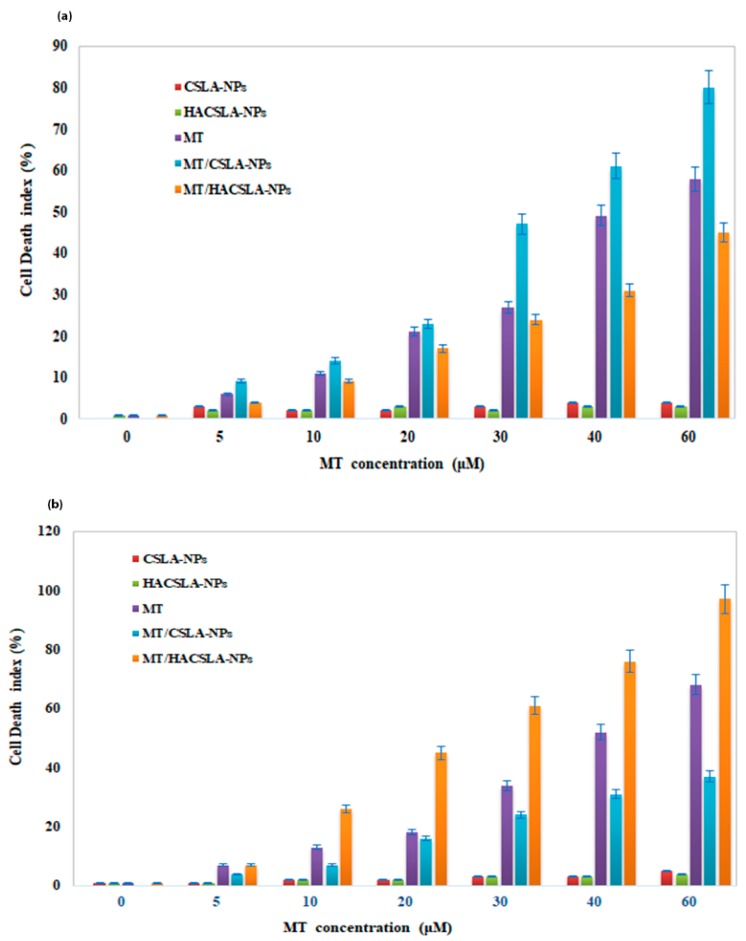
MT cytotoxicity measured with the Terminal deoxynucleotidyl transferase dUTP nick end labeling (TUNEL) assay in (**a**) MCF-7 and (**b**) BT-20 cells. All data represented by mean ± S.E.M (*p* < 0.05).

**Figure 9 molecules-25-01181-f009:**
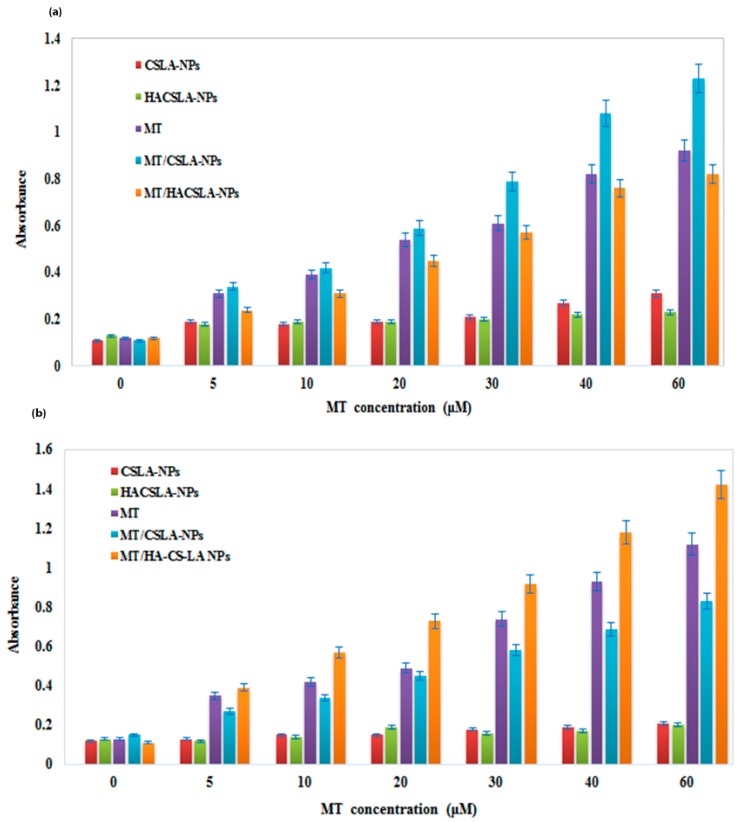
The activity of Caspase 3 was determined using the Caspase Colorimetric Assay Kit in (**a**) MCF-7 and (**b**) BT-20 cells. All data represented by mean ± S.E.M (*p* < 0.05).

**Figure 10 molecules-25-01181-f010:**
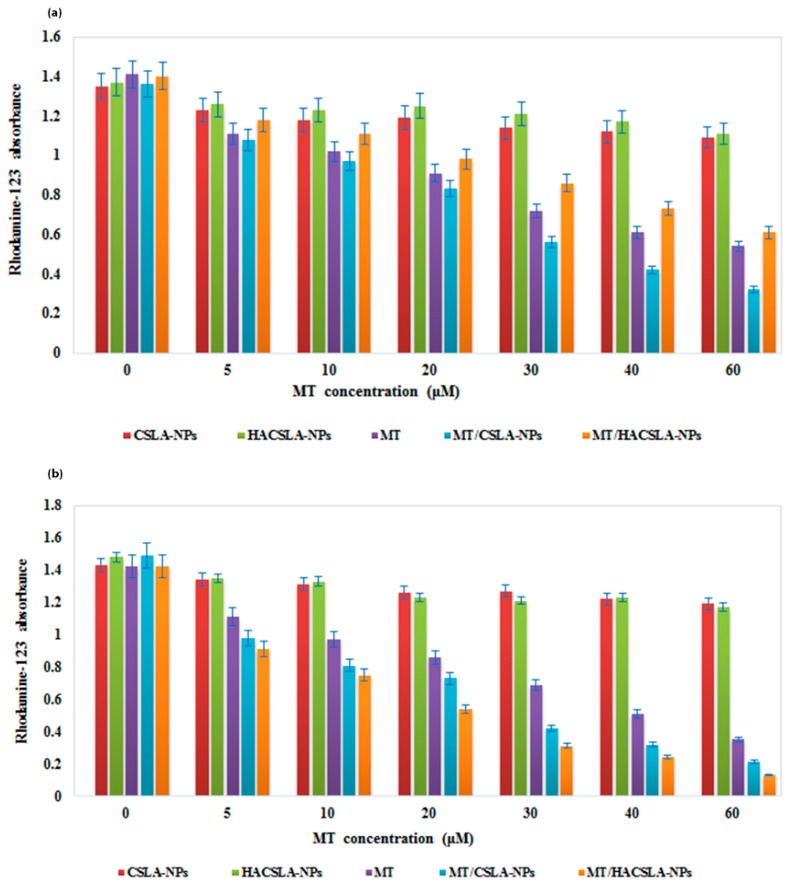
The effects of different treatments on the cell death on (**a**) MCF-7 and (**b**) BT-20 cells. All data represented by mean ± S.E.M. (*p* < 0.05).

**Figure 11 molecules-25-01181-f011:**
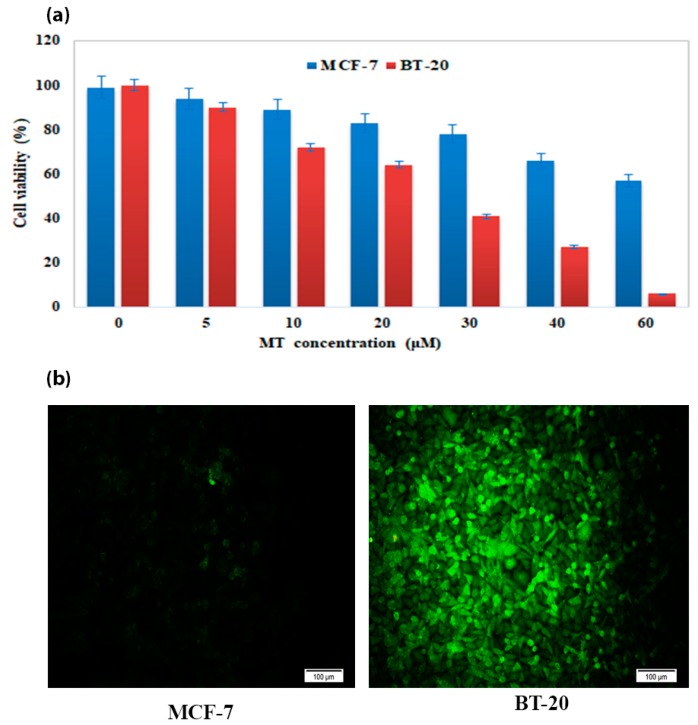
(**a**)The effects of different MT concentrations (0.0–60 μM) on the cell viability on MCF-7 (negative CD44) and BT-20 (positive CD44) cells, (**b**) Immunocytochemical staining of CD44.Very high expression was observed in BT-20 a breast cancer cell line, while the CD44 expression was not detectable in MCF-7.

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
