# Peer review of "Redox-Sensitive and Hyaluronic Acid-Functionalized Nanoparticles for Improving Breast Cancer Treatment by Cytoplasmic 17α-Methyltestosterone Delivery"

_molecules, 2020, doi:10.3390/molecules25051181_

Round 1
Reviewer 1 Report
The manuscript titled “Redox-sensitive and hyaluronic acid functionalized nanoparticles for improving breast cancer treatment by cytoplasmic 17α-Methyltestosterone delivery” by Rezaei et al propose a new method for breast cancer treatment by means of targeted drug delivery using reduction-responsive NPs.
Overall, I find that the manuscript is well-organized. The authors reported a detailed description of the synthesis of the reduction responsive NPs and have characterized the NPs structure and chemical characteristics. Furthermore, the authors have performed a proof of concept in vitro experiments to evaluate the feasibility of the proposed method. To improve the quality of the manuscript to make it suitable for publication in the Journal of Molecules, I suggest the following modifications:
1. Please proofread the manuscript for spelling and grammar.
2. Please improve all figure quality by increasing the resolution and use the same format and font for all figure x- and y-axes labels.
3. In section 3.6, please provide the EDX equipment model and manufacturer.
4. In section 3.7, please provide details on the XRD measurement conditions (i.e., X-ray source and wavelength, 2-theta range, filters and detectors used, etc).
5. In section 3.8, please specify the machine manufacturer used for UV-vis spectrophotometry.
6. In section 3.10, please provide details on what statistical methods are used.
Reviewer 2 Report
The paper is devoted to the creation of new redox-sensitive hyaluronic acid-chitosan-lipoic acid nanoparticles (HACLA - NPs) and hyaluronic acid functionalized nanoparticles (HA-NPs) for delivery and improving breast cancer treatment by cytoplasmic 17α-Methyltestosterone. The comparative study was made for the in vitro performance of the HA-coated and uncoated NPs. HA-coated NPs reduced MT release at pH 7.4 and increased MT release at pH 6.5. Therefore, HACSLA-NPs improved selective targeting against cancer cells by a faster drug release profile at pH 6.5. This simulates cytoplasm environment of the targeted tissue with an improved therapeutic efficacy. The authors confirmed highly efficient and targeted therapy of breast cancer cell lines (BT-20 and MCF-7).
Thus, the paper presents some new fundamental and practically interesting results, proved that HA surface modification played a major role in the biological responses, and these reduction-sensitive NPs can be utilized effectively in biomedical practice, The NPs obtained were characterized by the combination of different powerful physico-chemical methods.The text is well written and is logically presented.
At the same time some questions should be addressed to the authors:
- In Fig.1 the data of IR-spectroscopy study are presented for the different systems obtained, it seems to be suitable to mark by any way (by stars or numbers) the peaks, which are discussed in the text of presented paper/
- In Fig. 3 are presented the data on SEM study and energy dispersive X-ray analysis of the samples obtained. But there is no the presentation of the size distribution of the NPs ( in form of histogram or as distribution function), at the same time the title of this figure includes the words "size distribution"
The paper can be published in present form after minor revisions.
